# Teaching medical ethics and medical professionalism in Saudi public and private medical schools

**Mohammed AlRukban[1]\*, Fahad Alajlan[2], Ali Alnasser[3], Hisham Almousa[3], Sulaiman Alzomia[3], Abdullah Almushawah[3]**

1 Department of Family and Community Medicine, College of Medicine, King Saud University, Riyadh, Saudi Arabia, 2 Department of Family and Community Medicine, King Saud University Medical City, Riyadh, Saudi Arabia, 3 College of Medicine, King Saud University, Riyadh, Saudi Arabia, Saudi Arabia

\* mrukban@ksu.edu.sa

## Abstract

Medical ethics and professionalism are two essential parts of building up the identity of a competent physician. This study was conducted to determine the nature, content, and methods of medical ethics and professionalism education in Saudi public and private medical schools. It also sought to identify the challenges and obstacles in teaching and assessing medical ethics and professionalism and suggest appropriate changes. A cross-sectional study was carried out in Saudi private and public medical schools. To achieve the study's aim, an assessment tool in the form of a novel self-administered questionnaire was developed, piloted, and then used. A representative from each of the 28 Saudi medical schools participated in the study. Twenty-four (82.1%) responding medical schools have no medical ethics department. Most of the medical schools (64.2%) have 25% or less of their faculty staff who teach ethics holding a qualification in medical ethics. Most schools have a specific course for medical ethics and professionalism (85.7% and 57.1%, respectively). Multiple-choice questioning is the most popular assessment method in medical ethics and professionalism courses (89.3% and 60.7%, respectively). The need for more qualified staff and clear guidelines/resources is a significant drawback to the teaching of medical ethics. Therefore, the study recommends developing national guidelines dedicated to the undergraduate teaching curriculum from which courses would be designed to enhance medical ethics and medical professionalism.

## Introduction

Medical ethics is defined as a group of moral principles that guide healthcare professionals in the practice of medicine [1]. The teaching of medical ethics can be traced back to 2500 years ago. For example, the Hippocratic Oath highlighted the relationship between medicine and ethics during ancient times [2]. In Islamic culture, the Holy Quran references the values and ethical standards for Muslims' daily lives, including medical practice. These historical

**Data availability statement:** All relevant data are within the manuscript and its Supporting Information files.

**Funding:** The author(s) received no specific funding for this work.

**Competing interests:** The authors have declared that no competing interests exist.

documents' primary precept is "avoiding harm to patients" [3]. Moreover, cultures, traditions, and social morality have shaped and guided the development of ethical standards in the medical profession.

On the other hand, medical professionalism has different definitions in the literature. For example, a report produced in 2005 by the Royal College of Physicians in the UK delineated professionalism as "a set of values, behaviours, and relationships that underpin the trust the public has in doctors" [4]. Similarly, in 1995, The American Board of Internal Medicine concluded that professionalism "aspires to altruism, accountability, excellence, duty, honor, and respect for others" [5].

In previous years, several efforts have been made to create standards for medical ethics and professionalism worldwide. For instance, in 1995, the Association of American Medical Colleges published the Assessment of Professionalism Project, which provided a resource that defines medical professionalism for healthcare professionals [6]. Furthermore, on 9 October 2009, the United Nations Educational, Scientific, and Cultural Organization (UNESCO) created an internal ethics office that would advise the UNESCO membership about ethical values and standards of practice to mirror its "core values of integrity, professionalism, and respect for diversity [7].

Medical ethics and professionalism are two essential parts of building up the identity of a good physician. Medical education aims to produce knowledgeable, skilful, empathetic, and qualified physicians who will help promote public health and serve the community and the nation. Medical schools put enormous efforts into building up physicians' personal values and humanistic characteristics. In this context, medical ethics teaching became a global focus of the health sciences curricula in the previous three decades. The rising demand for medical ethics is due to significant advancements in the medical sciences, societal changes, public awareness, and increased rights movements. Although undergraduate medical curricula have widely included medical ethics and professionalism, the purpose, content, outcomes, evaluation and assessment, and teaching methods must be clarified [8].

Medical ethics courses are an essential part of the medical curricula for medical students and physicians in different parts of Saudi Arabia. According to one study, the total mean of medical students' agreement on the application of medical ethics in Hail was 80.1% [9]. In addition, this study found that 92.3% of physicians in government hospitals in Hail agreed on the importance of teaching medical ethics in medical training. On the other hand, another study revealed a deficit in medical ethics knowledge among physicians in Taif [10].

Several factors necessitate reassessing and defining medical ethics and professionalism courses in the undergraduate medical curriculum in Saudi Arabia. First, there needs to be more studies that review the teaching, evaluation, and content of medical ethics and professionalism courses in Saudi Arabia. There is also a need for a novel approach to teaching medical ethics and professionalism [11, 12]. Therefore, this study aimed to determine the nature, content, and methods of teaching and assessing medical ethics and professionalism in Saudi medical schools. It also attempted to identify the challenges and obstacles and to suggest appropriate solutions.

## Methods

This cross-sectional study involved the evaluation of the teaching and assessment of medical ethics and professionalism in public and private medical schools in Saudi Arabia. An assessment tool in the form of an electronic questionnaire was designed and sent to all medical schools in Saudi Arabia. An informed written consent was included on the first page of the electronic questionnaire. The online questionnaire was designed specifically for this study and

had three major sections. The first section provided a brief introduction about the research and its aims and informed consent at the end to assure confidentiality, while the remaining two sections were designed to gather data about medical ethics and professionalism education. The data that were collected included the sociodemographic status of the medical school, the total number of medical ethics and professionalism courses credit hours that it offered, whether these courses were mandatory or elective, commonly used teaching methods, the availability of teaching staff, the topics taught in the curriculum, common assessment methods used to assess students, and the obstacles faced while teaching the courses.

Medical schools were contacted by email or phone and asked to participate in the study. The person supervising the course(s) completed the questionnaire at each school. Data collection was carried out between September 2021 and February 2022.

### Statistical analysis

Data were automatically entered into Excel and were analyzed using SPSS, version 26. A $p$-value less than .05 was considered statistically significant.

### Ethical considerations

The Institutional Review Board (E-21-6075) at King Saud University in Riyadh, Saudi Arabia reviewed and approved this study. The confidentiality of the participants was preserved, as only the authors had access to the data.

## Results

Twenty-eight out of 33 (85%) medical schools in Saudi Arabia participated in this study. Nineteen (68%) were public medical schools, and nine (32%) were private. The date of establishment of the medical schools that participated differs widely. The College of Medicine at King Saud University, established in 1967, is the oldest, while the Fakeeh College of Medical Sciences and the Al-Rayan Medical College, both established in 2017, are the newest. Half of the medical schools are in Riyadh and Jeddah, which are the most populated cities, while the other half are from the provinces of Makkah, Qassim, Madinah, Sharqiyah, Hail, Jouf, Asir, Al-Baha, Najran, and Jazan. Table 1 displays the demographics of the participating medical schools. Of the medical schools participating in the study, 24 (82.1%) have no medical ethics department, while 4 (17.9%) have a medical ethics department. In 64% of the medical schools, 25% or less of the faculty who teach a medical ethics course hold a qualification in medical ethics. On the other hand, in 21.4% of the medical schools, more than 25% of the faculty who teach a medical ethics course hold a qualification in medical ethics.

The methods by which course content related to medical ethics and professionalism is taught within the curriculum are shown in Table 2. Eighty-five per cent of the medical schools have either a specific course for medical ethics only or have a specific course plus ethical content embedded into different courses, while 57.1% have either a specific course for medical professionalism only or both a specific course and professional content embedded into other courses. Contrastingly, 14.3% of medical schools do not have any specific course for medical ethics and 42.9% do not have a specific course for professionalism.

All medical schools with courses in medical ethics or professionalism make medical ethics and professionalism courses compulsory for students. Most medical schools (64.3%) have only one medical ethics course, while 21.4% have more than one course within their curriculum. On the other hand, 46.4% of medical schools have only one medical professionalism course.

Half of the medical ethics courses are given in the third and fourth year of the medical curriculum, while 42.9% of professionalism courses are taught in the second year. The number

**Table 1. Demographics of participating medical schools.**

| Region | Name | Type of medical school | Year of establishment |
|---|---|---|---|
| **Central** | King Saud University | Public | 1967 |
| | King Saud bin Abdulaziz University for Health Sciences | Public | 2005 |
| | Imam Mohammed bin Saud University | Public | 2007 |
| | Princess Norah bin Abdulaziz University | Public | 2012 |
| | Al Qassim University | Public | 2000 |
| | Shaqra University | Public | 2011 |
| | Majmah University | Public | 2010 |
| | Alfaisal University | Private | 2008 |
| | Dar Al Uloom University | Private | 2013 |
| | Vision Colleges | Private | 2014 |
| | Al Maarefa University | Private | 2009 |
| | Sulaiman Alrajhi University | Private | 2009 |
| **Western** | King Saud bin Abdulaziz University for Health Sciences | Public | 2005 |
| | Umm Al-Qura University | Public | 1996 |
| | Taibah University | Public | 2001 |
| | Ibn Sina National College for Medical Studies | Private | 2004 |
| | Batterjee Medical College | Private | 2005 |
| | Fakeeh College for Medical Sciences | Private | 2017 |
| | Al-Rayan Medical Colleges | Private | 2017 |
| **Eastern** | King Faisal University | Public | 2000 |
| | Imam Abdulrahman bin Faisal University | Public | 1975 |
| **Northern** | University of Hail | Public | 2008 |
| | Jouf University | Public | 2007 |
| **Southern** | King Khalid University | Public | 1980 |
| | Al-Baha University | Public | 2006 |
| | Najran University | Public | 2008 |
| | Jazan University | Public | 2006 |
| | University of Bisha | Public | 2014 |

of credit hours allocated for medical ethics and professionalism varies within medical schools. Most medical schools (67.8%) allocate 2–3 credit hours to a medical ethics course, while 32.1% allocate 2 credit hours or less to a professionalism course.

The study shows a strong correlation between the age of the medical school and the distribution of the contents of medical ethics ($p = .03$), with older schools only having ethical content embedded in other courses.

The most common topics covered in medical ethics courses were found to be abortion and the doctor-patient relationship (89.3%), while the most common topics included in professionalism courses were identified as breaking bad news and concepts and principles of professionalism (57.1%). The topics least covered in medical ethics courses were HIV/AIDS and immunizations (28.6%) while volunteering and community commitment (25%) were identified as the least-covered topics in professionalism courses. Table 3 summarizes the topics included in these two courses and the degree to which they are covered.

Table 4 shows the teaching and assessment methods of medical ethics and professionalism within the medical curriculum. The most popular teaching method used in medical ethics and professionalism courses is lecturing (82.1% and 57.1%, respectively). The employment of practical activities is the least commonly used instructional method in ethics and professionalism courses (15.9 and 14.3, respectively). The use of multiple-choice questions is the most

**Table 2. How the course content related to medical ethics or professionalism is implemented in the curriculum.**

| Item | Medical ethics | Professionalism |
|---|---|---|
| | N (%) | N (%) |
| **How contents related to medical ethics/professionalism are given** | | |
| Specific course and embedded content into different courses | 17 (60.7%) | 13 (46.4%) |
| Specific course | 7 (25%) | 3 (10.7%) |
| Content embedded into different courses but no specific course | 4 (14.3%) | 12 (42.9%) |
| **Is the course compulsory?** | | |
| Compulsory | 24 (85.7%) | 16 (57.1%) |
| Elective | 0 | 0 |
| No course | 4 (14.3%) | 12 (42.9%) |
| **Number of courses** | | |
| No course | 4 (14.3%) | 12 (42.9%) |
| 1 | 18 (64.3%) | 13 (46.4%) |
| 2 | 4 (14.3%) | 2 (7.1%) |
| 3 | 2 (7.1%) | 1 (3.6%) |
| **School years where courses are given** | | |
| 1st | 5 (15.6%) | 3 (14.3%) |
| 2nd | 5 (15.6%) | 9 (42.9%) |
| 3rd | 8 (25%) | 1 (4.8%) |
| 4th | 8 (25%) | 5 (23.8%) |
| 5th | 6 (16.8%) | 5 (23.8%) |
| **Number of credit hours** | | |
| No course | 4 (14.3%) | 12 (42.9%) |
| 1 | 0 | 2 (7.1%) |
| 2 | 9 (32.1%) | 7 (25%) |
| 3 | 10 (35.7%) | 4 (14.3%) |
| 4 | 2 (7.1%) | 1 (3.6%) |
| 5 | 0 | 1 (3.6%) |
| 6 | 2 (7.1%) | 1 (3.6%) |
| More than 8 | 1 (3.6%) | 0 |

popular assessment method used in medical ethics and professionalism courses (89.3% and 60.7%, respectively). Practical assessment is not commonly used in both courses, accounting for 3.6% in the ethics course and 0% in the professionalism course.

This study shows that the type of medical school (private or public) has no significant effect on teaching and assessing medical ethics and medical professionalism in Saudi medical schools ($p = 0.07$).

Most medical schools (78.6%) use the Saudi national ethics guidelines in developing their medical ethics curriculum. Benchmarking with international practice is the least resource used in developing a medical ethics curriculum (28.6%) [Fig 1].

The most common obstacles perceived in medical ethics and professionalism education are the lack of qualified staff (57.1% and 10.7%, respectively), the lack of guidelines/resources (28.6% and 7.1%, respectively), and the lack of student interest (25% and 7.1%, respectively).

## Discussion

The teaching of ethics in medical schools has developed tremendously over the last few decades, and medical ethics has become an essential component of medical curricula nationally and

**Table 3. Topics included in medical ethics and professionalism courses in Saudi medical schools' curriculum.**

| Ethics topics | N (%) | Professionalism topics | N (%) |
|---|---|---|---|
| Abortion | 25 (89.3%) | Breaking bad news | 16 (57.1%) |
| Doctor–patient relationship | 25 (89.3%) | Concepts and principles of professionalism | 16 (57.1%) |
| Brain death | 24 (85.7%) | Communication skills | 15 (53.6%) |
| Contraception and sterilization | 24 (85.7%) | Doctor-patient relationship | 15 (53.6%) |
| Cross-cultural issues and diverse beliefs | 24 (85.7%) | Interprofessional relationship | 13 (46.4%) |
| Confidentiality and privacy | 23 (82.1%) | Unprofessional behavior | 12 (42.9%) |
| Ethical principles | 23 (82.1%) | Doctor's character | 11 (39.3%) |
| Informed consent | 23 (82.1%) | Examining patients | 11 (39.3%) |
| Medical errors | 23 (82.1%) | Teamwork | 11 (39.3%) |
| Patient rights | 23 (82.1%) | Time management | 9 (32.1%) |
| Research ethics | 23 (82.1%) | Management and leadership | 8 (28.6%) |
| Communication with medical industry | 22 (78.6%) | Stress management | 8 (28.6%) |
| End-of-life issues | 22 (78.6%) | Volunteering and community commitment | 7 (25.0%) |
| Discrimination in healthcare | 21 (75.0%) | | |
| Doctor's rights | 20 (71.4%) | | |
| Individual autonomy | 20 (71.4%) | | |
| Organ transplantation | 20 (71.4%) | | |
| Cosmetic surgery | 18 (64.3%) | | |
| Terminal care | 18 (64.3%) | | |
| Saudi medico-legal system | 17 (60.7%) | | |
| History of ethics | 16 (57.1%) | | |
| Fiqh of patient prayer and fasting | 15 (53.6%) | | |
| Ethics of interviewing | 14 (50.0%) | | |
| Equity and equality | 14 (50.0%) | | |
| Medical necessity-assisted reproduction | 12 (42.9%) | | |
| Prophetic medicine | 10 (35.7%) | | |
| Genetics | 9 (32.1%) | | |
| HIV/AIDS | 8 (28.6%) | | |
| Immunizations | 8 (28.6%) | | |

internationally. Previous studies have demonstrated an increase in the importance of medical ethics education, which is supported by the findings of this study. In the UK, ethics curriculum development started officially with the "Pond Report" in 1987, which recommended ethics education development [13]. Afterwards, a report under the name "Tomorrow's Doctors" in 1993, prepared by the General Medical Council, contained a recommendation to include "ethics and legal issues relevant to the practice of medicine" in medical curricula [14]. Later, in 1998, the UK consensus statement listed the core contents of ethics to be included in undergraduate courses for medical students and recommended the consideration of ethics and law in clinical teaching [15]. Similarly, in the USA, the Association of American Medical Colleges reported medical educators' increasing agreement on the need to include ethics education as part of a medical school's curriculum [16]. Locally, medical ethics instructors within Saudi medical schools have exhibited a growing interest and increased commitment to teaching medical ethics to their students.

This study has found that all Saudi medical schools consider teaching medical ethics a cornerstone component in the curriculum and have made learning medical ethics compulsory for medical students. Also, these medical schools are increasingly interested in adding ethics courses to their curricula. This study also found that 85.7% of Saudi medical schools dedicate

**Table 4. Teaching and assessment methods of medical ethics and professionalism curriculum in medical schools.**

| Teaching methods | Medical ethics | Professionalism |
|---|---|---|
| | N (%) | N (%) |
| Lectures | 23 (82.1%) | 16 (57.1%) |
| Case studies | 20 (71.4%) | 8 (28.6%) |
| Student presentations | 12 (42.9%) | 6 (21.4%) |
| Seminars/tutorials | 12 (42.9%) | 5 (17.9%) |
| PBL | 7 (25%) | 6 (21.4%) |
| Practical activities | 5 (17.9%) | 4 (14.3%) |
| Assessment methods | | |
| MCQs | 25 (89.3%) | 17 (60.7%) |
| Assignments | 15 (53.6%) | 8 (28.6%) |
| SAQs | 12 (42.9%) | 8 (28.6%) |
| Research (project) | 4 (14.3%) | 2 (7.1%) |
| OSCEs | 2 (7.1%) | 3 (10.7%) |
| Video and online campaigns | 1 (3.6%) | 0 |
| Written essays | 0 | 1 (3.6%) |

a specific course for medical ethics compared to a similar study conducted in 2013, which showed that only 35.7% of medical schools had ethics taught as an independent course [17]. Indeed, the development of medical ethics teaching primarily depends on medical schools' willingness to devote more curricular time and efforts to achieve high standards in medical ethics education [12].

In this study, we investigated the relationship between the age of established medical school and the distribution of contents related to medical ethics. We found a statistically significant correlation between the age of the medical school and the establishment of separate medical ethics courses, as older medical schools tend to have ethical content embedded into other courses while newer medical schools are more likely to have a separate medical ethics course in their curriculum. When we compared our research to the literature, no prior studies were found to directly address the relationship between school age and ethics courses; however, some articles suggested a link between the two. For example, a study published in the UK in 1987 found that medical ethics education was integrated into several medical subjects, such as psychiatry, obstetrics and gynaecology, medicine, and general practice, rather than being offered as a distinct course [18]. The distribution of ethics content appears to be changing, as evidenced by a 1990 UK study where all participating medical schools had a dedicated medical ethics course [13]. Similarly, data from the United States indicates a significant increase in the implementation of dedicated medical ethics courses within medical school curricula. Only 4% of schools offered separate medical ethics courses in 1972, which increased to 34% by 1989 and reached 100% by 1994 [19].

The development of the medical ethics curriculum has evolved over the last five decades, both in the Kingdom of Saudi Arabia and other parts of the world. A study published in the United States in 1994 reported no global guideline for the ethics curriculum; instead, it was primarily based on cultural values and aspects [20]. This issue was also reported as part of a systematic review conducted in 2000, which opined a tremendous need for implanting a unified curriculum for teaching medical ethics [21]. In the Kingdom of Saudi Arabia, the results are similar. A study published in 2003 indicated that agreeing on specific content represents a challenge for teaching ethics at medical schools [22]. However, this study demonstrated that

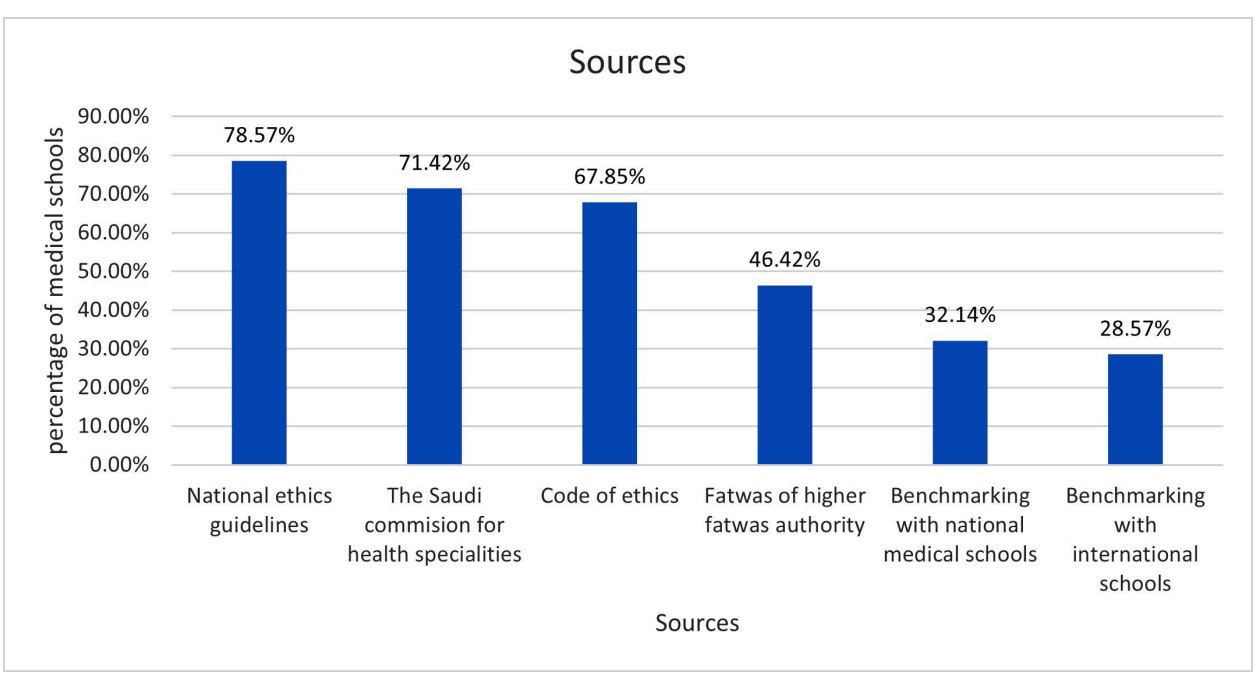

**Fig 1. Distribution of resources used in developing medical ethics curricula.**

more than 78% of medical schools have developed their curriculum based on national ethics guidelines.

In contrast to the many studies that discuss the development, teaching, and assessment of medical ethics, fewer studies highlight the development, teaching, and assessment of medical professionalism. This difference could be attributed to the fact that medical professionalism is usually taught as an integral part of medical ethics in many medical schools. However, many studies have pointed out the importance of medical professionalism and the need to develop the desired professionalism curriculum for medical students. For example, a study published in 1999 found that the teaching of professionalism in US medical schools was not always adequate and, as a result, medical courses needed to provide more explicit learning experiences about professionalism for medical students [23]. Another study conducted in 2012 in Canada concluded that teaching medical professionalism is essential; such courses should be taught explicitly and continuously, and a passing grade in these courses should serve as a prerequisite to advancement to the next level in the curriculum [24]. This study reported that Saudi medical schools emphasized medical ethics teaching more than medical professionalism. Most Saudi medical schools (85.7%) have either a specific course only or a specific course and embedded content into different courses for medical ethics, while 57.1% have either a specific course only or a specific course and embedded content into different courses for medical professionalism.

There are some discrepancies in determining the year in which the medical ethics and professionalism curricula should be offered nationally and internationally. In Europe, 45% of medical schools have medical ethics taught in clinical and preclinical years, 35% include it only in preclinical years, and 20% in clinical years [25]. Moodley in South Africa [26] and Miyasaka in Asia [27] have proposed the teaching of medical ethics during clinical and preclinical years. Locally, this study shows that 68.7% of Saudi medical schools tend to teach medical ethics in clinical years, half of the medical schools teach professionalism in clinical

years, and only 10.7% of them have medical ethics courses offered in both clinical and preclinical years.

This study showed that the type of medical school (private or public) has no significant effect on teaching medical ethics and professionalism in Saudi medical schools in the presence of courses, credit hours, teaching, and assessment methods. This finding complements those identified in a study done among American and Canadian medical schools in 2004, which reported a lack of relationship between the type of medical school (public or private) and the presence of compulsory medical ethics courses [12]. Additionally, a study of medical ethics teaching in Turkey in 2020 showed no significant difference between the public and private Turkish medical schools regarding medical ethics departments (68% and 75%, respectively) [28]. On the other hand, a study done in 2015 in Spain reported that private medical schools allocated 6 credit hours to medical ethics, while public medical schools only allocated 2 credit hours [29].

In this study, lecturing was the most frequently used method for teaching medical ethics and professionalism (82.1% and 57.1%, respectively), followed by case studies and student presentations. The methods of teaching medical ethics seem similar to the results of the previous study conducted in 2013, which also found that lecturing was the most common method of teaching medical ethics (64.3%), followed by case studies and then PBL sessions [1]. In contrast, students at King Abdulaziz University in Jeddah viewed using a portfolio workbook in ethics and professionalism courses as a useful tool for discussing ethical issues [30]. Moreover, a study done in Turkey found that only 30.4% of medical schools used lectures as the primary method of teaching medical ethics [28]. The literature recommends using student-centered approaches, such as case studies and small group discussions, for use in teaching medical ethics and professionalism over traditional methods [31, 32]. The researchers who conducted these studies explained that students learn social sciences more efficiently when finding the solutions to specific medical problems [31, 32].

This study also showed that multiple-choice questions are the most commonly used method for assessing students' achievement in medical ethics and professionalism, followed by assignments and short-answer questions. These results can be compared to other research showing assignments and workbook assessment exercises to be the most effective assessment methods within ethics courses [33]. Furthermore, Mattick and Bligh [34] found that essays and objective-structured clinical examinations are the preferred ways of assessment for medical ethics used in UK medical schools. The literature supports using clinical cases, such as those that students will face as physicians, to assess students' competency in medical ethics and professionalism [35].

The use of traditional methods in teaching and assessing medical ethics has both advantages and drawbacks. One of its advantages is that it offers a structured and organized way to present and evaluate information. Another advantage is that traditional methods provide a consistent framework for both educators and students. This can be particularly important in a field like medical ethics, where a clear understanding of ethical principles is crucial.

Moreover, traditional assessment methods such as written exams can provide objective metrics for evaluating students' comprehension of ethical principles. This objectivity is crucial when assessing competencies in a subject as nuanced as medical ethics. Despite these advantages, there are drawbacks to traditional methods. For instance, they may not fully engage students in the intricacies of ethical decision-making. Interactive and experiential learning, which is often lacking in traditional approaches, is vital for understanding the real-world applications of medical ethics.

This study showed that lack of qualified staff is the biggest obstacle facing teaching medical ethics and professionalism, followed by the lack of guidelines/resources and student interest. Besides, only 17.9% of the medical schools reported having a medical ethics department. The

lack of a medical ethics department could be attributed to the need for qualified staff. Some of these obstacles were also reported in a previous study done in American and Canadian medical schools, where it was found that the obstacles to teaching medical ethics included a lack of curricular focus on ethics, constraints on faculty time available for the preparation of instructional material on this topic, and the small percentage of faculty qualified to teach medical ethics [12].

## Conclusion

This study comprehensively evaluates the status of medical ethics and professionalism education across medical schools in Saudi Arabia. The findings highlight a growing commitment to integrating ethics education into medical curricula, with all participating schools deeming medical ethics a fundamental and compulsory component of medical education. Notably, there has been an encouraging increase in the allocation of specific courses dedicated to medical ethics, signalling a positive shift compared to previous years. However, a significant correlation between the age of medical schools and the distribution of medical ethics content suggests the need for further development in newer institutions. While the study highlights the existing use of traditional teaching methods, such as lectures, a shift towards more interactive student-centred approaches, as recommended by existing literature, could enhance the effectiveness of medical ethics education. Moreover, the lack of practical assessment methods in medical ethics and professionalism courses and the dependency on multiple-choice questions as an assessment method can be improved by using better assessment tools, such as clinical cases, giving a more valid evaluation of students' competencies in both domains.

## Limitations

There are some limitations observed throughout the study. The survey used to assess the curriculum solely targeted the program directors without reflecting on the academic staff and students' satisfaction or opinions about the courses. It would also be a great advantage if the curriculum were evaluated through independent evaluators rather than depending on the college self-assessment. Another limitation of this study is the lack of a standardized assessment tool or clear guidelines for assessing the level of performance.

## Supporting information

**S1 File.  Questionnaire.**
(PDF)

## Acknowledgments

Special thanks to the College of Medicine Research Center, Deanship of Scientific Research, King Saud University Riyadh, Saudi Arabia for their support.

We thank and acknowledge the deans and teaching staff in the participating medical schools.

## Author contributions

**Data curation:** Fahad Alajlan, Ali Alnasser, Hisham Almousa.

**Methodology:** Fahad Alajlan, Ali Alnasser, Abdullah Almushawah.

**Project administration:** Abdullah Almushawah.

**Visualization:** Fahad Alajlan.

**Writing – original draft:** Mohammed AlRukban, Fahad Alajlan, Ali Alnasser, Hisham Almousa, Sulaiman Alzomia, Abdullah Almushawah.

**Writing – review & editing:** Mohammed AlRukban, Ali Alnasser, Hisham Almousa, Sulaiman Alzomia, Abdullah Almushawah.

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
