## [Decision Letter · Decision Letter 0]

21 Nov 2023

PONE-D-23-32418Teaching Medical Ethics and Medical Professionalism in Saudi Public and Private Medical SchoolsPLOS ONE

Dear Dr. alzomia,

Thank you for submitting your manuscript to PLOS ONE. After careful consideration, we feel that it has merit but does not fully meet PLOS ONE’s publication criteria as it currently stands. Therefore, we invite you to submit a revised version of the manuscript that addresses the points raised during the review process.

We look forward to receiving your revised manuscript.

Kind regards,

Mukhtiar Baig, Ph.D.

Academic Editor

PLOS ONE

Journal Requirements:

3. In your Data Availability statement, you have not specified where the minimal data set underlying the results described in your manuscript can be found. PLOS defines a study's minimal data set as the underlying data used to reach the conclusions drawn in the manuscript and any additional data required to replicate the reported study findings in their entirety. All PLOS journals require that the minimal data set be made fully available. For more information about our data policy, please see http://journals.plos.org/plosone/s/data-availability .

"Upon re-submitting your revised manuscript, please upload your study’s minimal underlying data set as either Supporting Information files or to a stable, public repository and include the relevant URLs, DOIs, or accession numbers within your revised cover letter. For a list of acceptable repositories, please see http://journals.plos.org/plosone/s/data-availability#loc-recommended-repositories . Any potentially identifying patient information must be fully anonymized.

Important: If there are ethical or legal restrictions to sharing your data publicly, please explain these restrictions in detail. Please see our guidelines for more information on what we consider unacceptable restrictions to publicly sharing data: http://journals.plos.org/plosone/s/data-availability#loc-unacceptable-data-access-restrictions . Note that it is not acceptable for the authors to be the sole named individuals responsible for ensuring data access.

Reviewers' comments:

Reviewer's Responses to Questions

**Comments to the Author**

1. Is the manuscript technically sound, and do the data support the conclusions?

Reviewer #1: Partly

Reviewer #2: Yes

2. Has the statistical analysis been performed appropriately and rigorously?

Reviewer #1: No

Reviewer #2: Yes

3. Have the authors made all data underlying the findings in their manuscript fully available?

Reviewer #1: No

Reviewer #2: Yes

4. Is the manuscript presented in an intelligible fashion and written in standard English?

Reviewer #1: No

Reviewer #2: Yes

5. Review Comments to the Author

Reviewer #1: Dear Authors,

Although your study presents the results of original research on medical ethics teaching at universities in Saudi Arabia, it needs to be greatly elaborated as a whole.

The paper has got to contain statistics, and other allied analyses to perform a high technical standard, and data and findings should be described in sufficient detail.

The discussion needs a substantial elaboration rather than a superficial comments from some of the papers in the allied literature.

The Limitations should be placed in a separate part not inserted in the Discussion in a vague manner as it is now.

It appears that there is no "Conclusion" part in the manuscript. This part should be added to your piece with in depth elucidation by supporting knowledge based on the data you have gleaned.

The article has not been drafted in an intelligible style, and a thorough revision by a native speaker in standard English is highly recommended. The survey (questionnaire) used in the study should be placed in the manuscript in order to make readers understand the questions asked, and explained in consistency with authors' analyses in line with standards for data availability.

The most significant deficiency of the paper seems that the questionnaire has been filled by the administrative staff at universities as stated by authors in the manuscript, the survey should have rather been administered by the ethics teaching staff themselves in order to have more accurate and more substantial data.

Kind regards.

Reviewer #2: Authors have explored an important component of undergraduate medical training and have provided the evidence regarding the current status, which is lacking required uniformity and is deficient in many aspects despite an apparent willingness and effort to enrich the curriculum and training in relation to medical ethics and professionalism

However, there are areas where the study/manuscript needs improvement

1. At the start of result it was mentioned that 28 medical schools participated in the study. But in the same paragraph while discussing presence of ethics departments the count is 24+5=29

Of the medical schools participating in the study, 24 (82.1%) had no medical ethics department, while 5 (17.9%) had a medical ethics department

2. From the results, it appears that ethics and professionalisms are taught either as a course or embedded in other courses or both but separately from each other which may be the case but surely these are also combined under a single course in few if not many medical colleges. Authors need to clarify about this

3. It will be good to mention the year in which the “Pond’s report” and “Tomorrow's Doctors,” were published

4. Authors have mentioned that, “The development of the medical ethics curriculum has evolved over the last century, both in the Kingdom of Saudi Arabia and in other parts of the world”. From the discussion, it is apparent that these developments belong to maximum last 50 years and not full century. It is suggested to replace the phrase ‘last century’ with ‘last 5 decades’

5. More importantly, there is a definite deficiency in literature review. The authors have mentioned about different teaching strategies and assessment tools being employed in medical schools of Saudi Arabia. For example, they mentioned that,

“There have been no changes in the assessment methods used in medical ethics courses since 2013, when it was reported that the use of MCQs was the most popular method used for assessing medical ethics”

Whereas, the following two articles (published in 2016 and 2021) related to teaching and assessing medical ethics and professionalism in a Saudi Arabian medical college are indicating innovations with successful use of “Portfolio Workbook”, both as a teaching modality and as a major component of assessment. It is recommended that authors should go through these papers and include the innovations in their discussion

a. Innovation in ethics and professionalism course: Early experience with portfolio-workbook. Shamim MS, Zubairi NA, Sayed MH, Gazzaz ZJ. J Pak Med Assoc. 2016 Sep;66(9):1149-1153. PMID: 27654737.

b. Systematic development and refinement of a contextually relevant strategy for undergraduate medical ethics education: a qualitative study. Shamim MS, Torda A, Baig LA, Zubairi N, Balasooriya C. BMC Med Educ. 2021 Jan 6;21(1):9. doi: 10.1186/s12909-020-02425-6. PMID: 33407410; PMCID: PMC7786930.

6. Finally, it is recommended that authors should emphasize more in their conclusion about lack of practical assessment tools as compared to dependency on MCQs

6. PLOS authors have the option to publish the peer review history of their article (what does this mean? ). If published, this will include your full peer review and any attached files.

**Do you want your identity to be public for this peer review?** For information about this choice, including consent withdrawal, please see our Privacy Policy .

Reviewer #1: No

Reviewer #2: **Yes: ** Nadeem Alam Zubairi

---

## [Author Response · Author response to Decision Letter 0]

14 Jan 2024

To,

The Editor, Reviewers,

PLOS ONE

Date: 14/1/2023

Subject: Addressing Editor/Reviewer comments

Dear Editor, Reviewers,

We express our thanks and gratitude for your valuable detailed review and comments. We appreciate your feedback which helped improve our manuscript. The authors tried their best to address all the concerns and comments the reviewers and the editor had. All the points from the reviewers’ were kept in original black, and our responses ( in red) were inserted under each of these points.

Responses to the editors:

Please ensure that your manuscript meets PLOS ONE's style requirements, including those for file naming. The PLOS ONE style templates.

We have reviewed the PLOS style requirements and ensured the manuscript met them.

We note that the grant information you provided in the ‘Funding Information’ and ‘Financial Disclosure’ sections do not match. When you resubmit, please ensure that you provide the correct grant numbers for the awards you received for your study in the ‘Funding Information’ section.

The study did not receive any financial funding. The support received from the Research center at King Saud University was for IRB approval. The acknowledgment to the research center at King Saud University was added under the acknowledgment section.

In your Data Availability statement, you have not specified where the minimal data set underlying the results described in your manuscript can be found. PLOS defines a study's minimal data set as the underlying data used to reach the conclusions drawn in the manuscript and any additional data required to replicate the reported study findings in their entirety. All PLOS journals require that the minimal data set be made fully available.

The authors declare approval of publishing this manuscript in the PLOS ONE journal in case of acceptance with no restriction to the data. All relevant data are within the paper and its supporting files.

The reference list was reviewed, and we found two references (number 10 and number 13) were retracted. These references were replaced by two other relevant references.

Response to Reviewer #1:

Although your study presents the results of original research on medical ethics teaching at universities in Saudi Arabia, it needs to be greatly elaborated as a whole.

The study aimed to evaluate the curriculum, content, and instructional methods employed in teaching medical ethics and professionalism within Saudi medical schools. Additionally, it sought to identify challenges and barriers, make comparisons with existing literature, and provide recommendations for improving the teaching of medical ethics and professionalism in Saudi medical schools. The research extensively examined the overall approach to teaching medical ethics, citing examples such as "This study reported that Saudi medical schools emphasized medical ethics teaching more than medical professionalism." Furthermore, it noted, "This study has found that all Saudi medical schools consider teaching medical ethics a cornerstone component in the curriculum and have made learning medical ethics compulsory for medical students." However, specific instances of the teaching of medical ethics in Saudi were also highlighted to support the authors' hypothesis.

The paper has got to contain statistics, and other allied analyses to perform a high technical standard, and data and findings should be described in sufficient detail.

This study primarily adopts a descriptive approach, utilizing statistical methods such as paired t-tests, Chi-square tests, frequencies, and measures of central tendency. Additionally, the study incorporates correlations, including an examination of the relationship between the age of medical schools and the distribution of medical ethics content (p-value=0.03). Notably, older schools were found to exclusively integrate ethical content into other courses.

The discussion needs substantial elaboration rather than superficial comments from some of the papers in the allied literature.

We carefully reviewed the discussion section and incorporated necessary changes in the edited version.

The Limitations should be placed in a separate part not inserted in the Discussion in a vague manner as it is now.It appears that there is no "Conclusion" part in the manuscript. This part should be added to your piece with in depth elucidation by supporting knowledge based on the data you have gleaned.

Well noted. A separate section for conclusion and limitations was created.

The article has not been drafted in an intelligible style, and a thorough revision by a native speaker in standard English is highly recommended.

The manuscript was sent for language editing by a native speaker. The language editing certification is included in the supplementary files.

The survey (questionnaire) used in the study should be placed in the manuscript to make readers understand the questions asked, and explained in consistency with authors' analyses in line with standards for data availability.

The survey is available in the supplementary files to be reviewed by readers.

The most significant deficiency of the paper seems that the questionnaire has been filled by the administrative staff at universities as stated by authors in the manuscript, the survey should have rather been administered by the ethics teaching staff themselves in order to have more accurate and more substantial data.Kind regards.

We concur that the completion of the questionnaire should be undertaken by the teaching staff, and this is precisely what occurred. The survey was filled out by the teaching staff overseeing the course, as explicitly stated in the methods section: "The person supervising the course completed the questionnaire in each medical school." In Saudi medical schools, course supervisors are tasked with the development, instruction, and assessment aspects.

Responses to Reviewer # 2:

Thank you Dr. Nadeem Alam Zubairi for your valuable comments.

1. At the start of the result it was mentioned that 28 medical schools participated in the study. But In the same paragraph while discussing presence of ethics departments the count is 24+5=29

This mistake was corrected in the result section as the following:

Of the medical schools participating in the study, 24 (85.7%) had no medical ethics department, while 4 (14.3 %) had a medical ethics department.

2. From the results, it appears that ethics and professionalism are taught either as a course or embedded in other courses or both but separately from each other which may be the case but surely these are also combined under a single course in a few if not many medical colleges. Authors need to clarify about this.

The study investigated whether Medical Professionalism/Medical Ethics was delivered as a distinct course, integrated into other courses, or a combination of both. Our data encompassed two disciplines, examining scenarios where these elements were either integrated or taught separately. The objective did not include exploring combined courses, and such inclusion would not impact the current study's results. However, examining combined courses could be a potential focus for future research.

3. It will be good to mention the year in which the “Pond’s report” and “Tomorrow's Doctors,” were published

We added the year in which the “Pond’s report” and “Tomorrow's Doctors,” were published in the edited version.

4. Authors have mentioned that, “The development of the medical ethics curriculum has evolved over the last century, both in the Kingdom of Saudi Arabia and in other parts of the world”. From the discussion, it is apparent that these developments belong to maximum last 50 years and not full century. It is suggested to replace the phrase ‘last century’ with ‘last 5 decades’

We have edited the point as recommended in the new version.

5. More importantly, there is a definite deficiency in the literature review. The authors have mentioned different teaching strategies and assessment tools being employed in medical schools in Saudi Arabia. For example, they mentioned that,

“There have been no changes in the assessment methods used in medical ethics courses since 2013, when it was reported that the use of MCQs was the most popular method used for assessing medical ethics”

Whereas, the following two articles (published in 2016 and 2021) related to teaching and assessing medical ethics and professionalism in a Sudia Arabian medical college are indicating innovations with successful use of “Portfolio Workbook”, both as a teaching modality and as a major component of assessment. It is recommended that authors should go through these papers and include the innovations in their discussion

a. Innovation in ethics and professionalism course: Early experience with portfolio-workbook. Shamim MS, Zubairi NA, Sayed MH, Gazzaz ZJ. J Pak Med Assoc. 2016 Sep;66(9):1149-1153. PMID: 27654737.

b. Systematic development and refinement of a contextually relevant strategy for undergraduate medical ethics education: a qualitative study. Shamim MS, Torda A, Baig LA, Zubairi N, Balasooriya C. BMC Med Educ. 2021 Jan 6;21(1):9. doi: 10.1186/s12909-020-02425-6. PMID: 33407410; PMCID: PMC7786930.

We added these references to our literature and included them in the discussion section.

6. Finally, it is recommended that authors should emphasize more in their conclusion about the lack of practical assessment tools as compared to dependency on MCQs

.

We have edited the conclusion and ensured to include the lack of practical assessment tools in the conclusion.

---

## [Decision Letter · Decision Letter 1]

29 Jan 2024

Teaching Medical Ethics and Medical Professionalism in Saudi Public and Private Medical Schools

PONE-D-23-32418R1

Dear Dr. Alzomia,

We’re pleased to inform you that your manuscript has been judged scientifically suitable for publication and will be formally accepted for publication once it meets all outstanding technical requirements.

An invoice for payment will follow shortly after the formal acceptance. To ensure an efficient process, please log into Editorial Manager at http://www.editorialmanager.com/pone/ , click the 'Update My Information' link at the top of the page, and double check that your user information is up-to-date. If you have any billing related questions, please contact our Author Billing department directly at authorbilling@plos.org .

If your institution or institutions have a press office, please notify them about your upcoming paper to help maximize its impact. If they’ll be preparing press materials, please inform our press team as soon as possible -- no later than 48 hours after receiving the formal acceptance. Your manuscript will remain under strict press embargo until 2 pm Eastern Time on the date of publication. For more information, please contact onepress@plos.org .

Kind regards,

Mukhtiar Baig, Ph.D.

Academic Editor

PLOS ONE

Reviewers' comments:

Reviewer's Responses to Questions

**Comments to the Author**

1. If the authors have adequately addressed your comments raised in a previous round of review and you feel that this manuscript is now acceptable for publication, you may indicate that here to bypass the “Comments to the Author” section, enter your conflict of interest statement in the “Confidential to Editor” section, and submit your "Accept" recommendation.

Reviewer #1: All comments have been addressed

Reviewer #2: All comments have been addressed

2. Is the manuscript technically sound, and do the data support the conclusions?

Reviewer #1: Yes

Reviewer #2: Yes

3. Has the statistical analysis been performed appropriately and rigorously?

Reviewer #1: Yes

Reviewer #2: Yes

4. Have the authors made all data underlying the findings in their manuscript fully available?

Reviewer #1: Yes

Reviewer #2: Yes

5. Is the manuscript presented in an intelligible fashion and written in standard English?

Reviewer #1: Yes

Reviewer #2: Yes

6. Review Comments to the Author

Reviewer #1: The revision fulfilled and presented by the authors is sufficient and plausible together with the bullet by bullet explanations and responses. The revised manuscript can be accepted for publication. Thanks.

Reviewer #2: Authors have made necessary corrections keeping in view the observations made by the reviewers and have included the latest related studies with the references

7. PLOS authors have the option to publish the peer review history of their article (what does this mean? ). If published, this will include your full peer review and any attached files.

**Do you want your identity to be public for this peer review?** For information about this choice, including consent withdrawal, please see our Privacy Policy .

Reviewer #1: No

Reviewer #2: No

---

## [Editor Report · Acceptance letter]

19 Feb 2024

PONE-D-23-32418R1

PLOS ONE

Dear Dr. Alzomia,

I'm pleased to inform you that your manuscript has been deemed suitable for publication in PLOS ONE. Congratulations! Your manuscript is now being handed over to our production team.

Lastly, if your institution or institutions have a press office, please let them know about your upcoming paper now to help maximize its impact. If they'll be preparing press materials, please inform our press team within the next 48 hours. Your manuscript will remain under strict press embargo until 2 pm Eastern Time on the date of publication. For more information, please contact onepress@plos.org .

If we can help with anything else, please email us at customercare@plos.org .

Kind regards,

on behalf of

Professor Mukhtiar Baig

Academic Editor

PLOS ONE